# Early post-partum viremia predicts long-term non-suppression of viral load in HIV-positive women on ART in Malawi: Implications for the elimination of infant transmission

**Megan Landes**[1,2]\*, **Monique van Lettow**[1,3], **Joep J. van Oosterhout**[1,4,5], **Erik Schouten**[6], **Andrew Auld**[7], **Thokozani Kalua**[8], **Andreas Jahn**[8,9], **Beth A. Tippett Barr**[10]

**1** Dignitas International, Zomba, Malawi, **2** Department of Family and Community Medicine, University of Toronto, Toronto, Canada, **3** Dalla Lana School of Public Health, University of Toronto, Toronto, Canada, **4** Partners in Hope, Lilongwe, Malawi, **5** David Geffen School of Medicine, University of California Los Angeles, Los Angeles, California, United States of America, **6** Management Sciences for Health, Lilongwe, Malawi, **7** Centers for Disease Control and Prevention, Lilongwe, Malawi, **8** Ministry of Health, Lilongwe, Malawi, **9** I-TECH, Department of Global Health, University of Washington, Seattle, Washington, United States of America, **10** Centers for Disease Control and Prevention, Kisumu, Kenya

\* mclandes@gmail.com

## Abstract

### Background

Long-term viral load (VL) suppression among HIV-positive, reproductive-aged women on ART is key to eliminating mother-to-child transmission (MTCT) but few data exist from sub-Saharan Africa. We report trends in post-partum VL in Malawian women on ART and factors associated with detectable VL up to 24 months post-partum.

### Methods

1–6 months post-partum mothers, screened HIV-positive at outpatient clinics in Malawi, were enrolled (2014–2016) with their infants. At enrollment, 12- and 24-months post-partum socio-demographic and PMTCT indicators were collected. Venous samples were collected for determination of maternal VL (limit of detection 40 copies/ml). Results were returned to clinics for routine management.

### Results

596/1281 (46.5%) women were retained in the study to 24 months. Those retained were older (p<0.01), had higher parity (p = 0.03) and more likely to have undetectable VL at enrollment than those lost to follow-up (80.0% vs 70.2%, p<0.01). Of 590 women on ART (median 30.1 months; inter-quartile range 26.8–61.3), 442 (74.9%) with complete VL data at 3 visits were included in further analysis. Prevalence of detectable VL at 12 and 24 months was higher among women with detectable VL at enrollment than among those with undetectable VL (74 detectable VL results/66 women vs. 19/359; p<0.001). In multivariable analysis (adjusted for age, parity, education, partner disclosure, timing of ART start and self-reported

**Data Availability Statement:** All relevant data are within the paper and its Supporting Information files.

**Funding:** This project has been supported by the President's Emergency Plan for AIDS Relief (PEPFAR) through the Centers for Disease Control and Prevention (CDC) under the terms of cooperative agreement U2GGH000721. CDC staff were involved as co-investigators, assisting in protocol development and approval and manuscript authorship. The authors acknowledge full access to all the data and final responsibility for submission. The findings and conclusions in this report are those of the authors and do not necessarily represent the official position of the funding agencies.

**Competing interests:** The authors have declared that no competing interests exist.

adherence), detectable VL at 24 months was 9 times more likely among women with 1 prior detectable VL (aOR 9.0; 95%CI 3.5–23.0, p<0.001) and 226 times more likely for women with 2 prior detectable VLs (aOR 226.4; 95%CI 73.0–701.8, p<0.001).

## Conclusions

Detectable virus early post-partum strongly increases risk of ongoing post-partum viremia. Due to high loss to follow-up, the true incidence of detectable VL over time is probably underestimated. These findings have implications for MTCT, as well as for the mothers, and call for intensified VL monitoring and targeted adherence support for women during pregnancy and post-partum.

## Introduction

In the last decade, prevention of maternal to child transmission of HIV (PMTCT) programs worldwide have transitioned from providing intermittent antiretroviral treatment (ART) for HIV-infected women during pregnancy, delivery and breastfeeding, to initiating lifelong ART (i.e., 'Option B+') [1]. This change has aligned PMTCT programs with evolving World Health Organization (WHO) universal test and treat guidelines for all persons living with HIV (i.e., 'Treat All') [2]. As a result, the number of women of reproductive age now initiating ART in sub-Saharan Africa has dramatically increased [3], and the benefit of long-term viral load suppression (VLS) during subsequent pregnancies, particularly in high fertility settings, has PMTCT outcomes now approaching those in high-income settings.

While pregnancy remains an important entry point for the current global 'Treat All' strategy, there is limited information regarding longitudinal VLS in this population [3]. Research studies reporting VLS estimates at various time points in the post-partum period signal particular challenges in reaching the UNAIDS goal of 90% VLS [4–7]. Further, among the few available population level estimates, we see a similar signal with Zimbabwe reporting VLS (<1000 copies/mL) in post-partum women on ART as 81.2% (CI: 79.4–83.1) at 4–12 weeks and 85.2% (CI: 82.9–87.4) at 12 months [8] and Uganda reporting a 3 year VLS (<1000 copies/mL) among post-partum women of 76% [9].

Several studies highlight difficulties in sustaining both adherence and VLS throughout the post-partum period. In Malawi, national routine program data demonstrate that only 30% women initiating ART during pregnancy or breastfeeding maintained adequate self-reported adherence at all visits over 2 years [10]. Data from South Africa show that during the first post-partum year, 30% of women on ART did not maintain VLS [11, 12], and in Zimbabwe, 50% of women did not maintain durable VLS over 12 months [8]. As detectable viremia determines risk of MTCT during pregnancy and breastfeeding [11, 13] as well as progression of illness in the mother, understanding and improving long-term VLS among women starting ART in pregnancy is key to eliminating MTCT and promoting the health of mothers.

In 2011, Malawi was the first country to implement 'Option B+' and has since shown large increases in women initiating ART in pregnancy [14], along with a remarkable reduction in early MTCT [15]. The National Evaluation of Malawi's PMTCT Program (NEMAPP) study was launched in 2014 to evaluate the effectiveness of Option B+ by enrolling HIV-infected women and their infants at 4–26 weeks post-partum and following them annually for 2 years [16]. Within this nationally representative cohort, uptake of PMTCT services was very high at the time of enrollment: 97.8% of women knew their HIV status, 96.3% of these were on ART,

and among a sub-set of these women 87.9% had achieved VLS (<1000 copies/ml) [13, 16]. Here we describe trends in detectable viral load (VL) among women on ART and retained until 24 months post-partum and we explore factors associated with VLS over time.

## Methods

This is a nested study of HIV-infected mothers presenting with their 1 to 6-month-old infants at outpatient clinics in Malawi, where they were enrolled for longitudinal follow-up in the NEMAPP study between October 2014 and March 2016 (with follow-up visits till March 2018). The study period started three years after the national implementation of 'Option B+' PMTCT guidelines which provided lifelong ART (i.e., tenofovir/lamivudine/efavirenz) for all pregnant and breastfeeding women [17]. At the time of the study, the national HIV program was in the early stages of implementing routine VL monitoring and coverage was still limited.

NEMAPP used a multistage cluster design to sample 54 sites across Malawi [15] to provide national representative 24-month outcomes of MTCT. The subset included in this study, based on regional strata, were enrolled for intensive clinical and laboratory monitoring at 13 health facilities across 8 districts. A sub-set sample of 1324 HIV-positive mothers was calculated to estimate VLS based on an estimated 50% suppression rate and 50% loss to follow-up for a precision of 2.5% with a 95% confidence interval (95% CI) and an assumed design effect of 2.0.

Women in selected sites were simultaneously consented for enrolment in the main study and in this subset for more in-depth clinical and laboratory monitoring. Guardian-infant pairs were excluded from this current analysis (Fig 1). Mother-infant pairs were followed up at 12 and 24 months post-partum, with observed window periods of 10 to 18 and 20 to 28 months post-partum, respectively.

At enrolment, 12 and 24 months, mothers were interviewed by trained health facility staff using structured (pre-tested) questionnaires to obtain socio-demographic information, HIV status at screening, disclosure to partner status, uptake and timing of PMTCT/ART, self-reported health status and adherence to treatment (as self-reported number of days of missed ART in the last month, with 'optimal adherence' defined as 0–1 days of missed ART). 'Durable adherence' at any timepoint is defined as 'optimal adherence' at all time points. When possible, mothers' health booklets and Ministry of Health registers were checked for accuracy of responses.

Maternal HIV VL testing was conducted on venous samples (Abbott Real-Time HIV-1 Assay, Abbott Laboratories, Chicago, IL) of all women regardless of ART status at enrollment, 12 and 24 months. VLS is defined as HIV 1-RNA <1000 copies/mL as per the Malawi national HIV guidelines [17]. We categorized VL results as 'undetectable' (<40 copies/mL) and 'detectable' (i.e., >40 copies/mL) with further sub-classification as 'low-detectable' (40–1000 copies/mL) and 'unsuppressed' (>1000 copies/mL). Further, we defined 'durably undetectable VL' as VL<40 copies/mL at all three study visits and 'persistently detectable VL' as VL>40 copies/mL at all three visits. The general term 'viremia' was used to describe the presence of any detectable virus in the blood.

Missing data were treated as additional categories. Crude percentages were calculated and comparisons between groups were made using chi-square tests for categorical variables and non-parametric tests for medians, using normal approximation (Wald) methods to calculate confidence intervals. Among women with complete VL observations, multivariable logistic regression analysis was used to identify characteristics associated with detectable vs. undetectable VL, with losing detectable VL vs. retaining undetectable VL, and with persistently detectable VL vs. those who had at least one undetectable VL. Univariate odds ratios (OR) with 95%

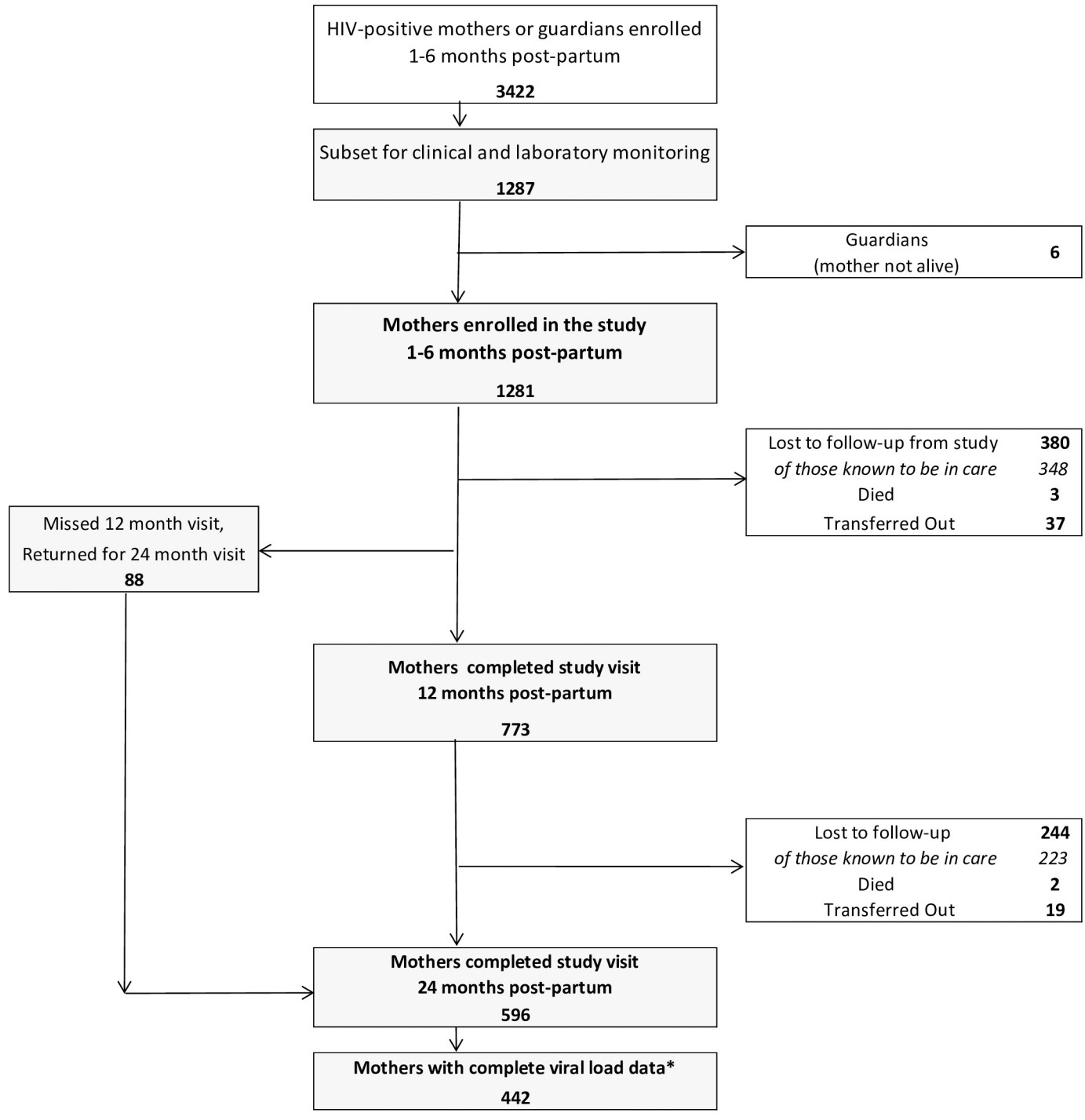

* 3 Viral load measures, at 1-6 month (enrolment), 12 and 24 months post-partum

**Fig 1. Mothers enrolled, followed and retained in the study up to 24 months post-partum.**

CI were calculated for each variable in the model using normal approximation (Wald) methods. Adjusted OR (aOR) with 95% CI were calculated for each model after adjustment for age, parity, education level, partner disclosure, timing of ART initiation, previous VL results and

adherence. All variables were simultaneously entered in the logistic regression model and tested for removal through backward stepwise selection. A 0.05 significance level was set for all statistical testing. Analyses were conducted using IBM SPSS Statistics 26 (IBM, Armonk, NY, USA).

Ethical approval was provided by Malawi's National Health Sciences Research Committee (#1262) and the University of Toronto (#30448). The US Centers for Disease Control and Prevention (CDC) reviewed and approved as research according to human research protection procedures (#2014-054-7), but was not engaged. All participants provided written informed consent.

## Results

Overall, 1281 HIV-infected mothers were enrolled in the study at 1–6 months post-partum with a median age of 29 years (interquartile range [IQR] 24–33) and parity of 3 (IQR 2–4) (data not shown). The majority of women (65.5%; n = 839) had either no formal or only primary level education. At enrollment, 94.3% (1208/1281) women already knew their HIV status and of these, 97.5% (1178/1208) were on ART. The median time on ART at enrollment was 8.3 months (IQR 5.2–39.5), with 47.6% (561/1178) of women having started ART pre-conception and 50.1% (590/1178) post-conception (Table 1).

Fig 1 describes follow-up and retention in the study. Overall, 773 (60.3%) women completed a 12 month study visit and 596 (46.5%) women completed a 24 month study visit, and of these, 97.7% (755/773) and 99.0% (590/596) were on ART, respectively (Table 2). Among women who did not complete the study, Ministry of Health registers confirmed that 91.5% (571/624) were known to be alive on ART at 24 months post-partum. In comparison to women alive and on ART but lost to follow-up from the study at 24 months, women remaining in the study at 24 months were more likely to be older (>30 years; p<0.01), have higher parity (p = 0.03) and more likely to be on ART at enrollment (p = 0.01). Further, women retained in the study had higher rates of undetectable VL at enrollment (VL <40 copies/mL) than those not (80.0% vs. 70.2%; p<0.01; S1 Table).

Overall, 86.7% (1021/1178) of women on ART had optimal adherence (i.e., 0–1 days missed ART in past month) at enrollment, 92.2% (696/755) at 12 months and 91.4% (539/590) at 24 months (Table 2). Further, 79.5% (600/755) of women retained had durable adherence at 12 months and 80.8% (477/590) had durable adherence at 24 months. Overall, among women with available VL data, VLS ratios (VL <1000 copies/mL) at enrollment, 12 and 24 months were 87.4% (1002/1147), 91.5% (658/719), 91.1% (492/540), respectively.

Table 3 describes VL trends among women with complete VL data over the 24 months post-partum period (N = 442), among whom 17 were not yet on ART at enrollment but subsequently started (10 were new diagnoses). Among women on ART at enrollment (n = 425), 80.7% (343/425) women had durable undetectable VL (<40 copies/mL) over 24 months, 11.8% (50/425) experienced at least one episode of viremia, and 7.5% (32/425) had persistently detectable VL. The prevalence of detectable VL at enrolment, 12 and 24 months were 15.5% (66/425), 9.9% (42/425) and 12.0% (51/425). The prevalence of detectable VL at 12 and 24 months was higher among women with detectable VL at enrollment than those with undetectable VL (74 detectable VL measures/66 women vs. 19/359; p<0.001).

Further, Fig 2 shows the results of sequential VL testing at 1–6, 12 and 24 months post-partum where plots are of VL trajectories for individual mothers stratified by ART start at either pre-conception, post-conception or post-enrollment. Among the women starting ART pre- and post-conception, 16.3% (33/203) and 22.1% (49/222; p = 0.13), respectively, had at least one episode of viremia within 24 months. Overall, women with newly diagnosed HIV at

**Table 1. Characteristics of HIV-infected mothers enrolled for follow-up to 24 months.**

| | Enrolment | Annual visit 1 | Annual visit 2 |
|---|---|---|---|
| | 1–6 months post-partum | 12 months post-partum | 24 months post-partum |
| N | 1281 | 773 | 596 |
| **Region where mother resides and attends health care** | | | |
| North Central Rural | 335 (26.2) | 186 (24.1) | 161 (27.0) |
| North Central Urban | 392 (30.6) | 193 (25.0) | 131 (22.0) |
| South Rural | 156 (12.2) | 123 (15.9) | 111 (18.6) |
| South Urban | 398 (31.1) | 271 (35.1) | 193 (32.4) |
| **Mothers' age in years** | | | |
| ≤19 | 79 (6.2) | 50 (6.5) | 35 (5.9) |
| 20–24 | 287 (22.4) | 145 (18 .8) | 102(17.1) |
| 25–29 | 335 (26.2) | 197 (25.5) | 152 (25.5) |
| ≥ 30 | 576 (45.0) | 379 (49.0) | 305 (51.2) |
| Missing | 4 (0.3) | 2 (0.3) | 2 (0.3) |
| **Parity** | | | |
| 1 | 192 (15.0) | 108 (14.0) | 76 (12.8) |
| 2–3 | 633 (49.4) | 366 (47.3) | 275 (46.1) |
| ≥ 4 | 454 (35.4) | 298 (38.6) | 244 (40.9) |
| Missing | 2 (0.2) | 1 (0.1) | 1 (0.2) |
| **Months post-partum at visit** | | | |
| 1–3 | 931 (72.7) | | |
| 4–6 | 350 (27.3) | | |
| 10–16 | | 713 (92.2) | |
| 17–18 | | 60 (7.8) | |
| 20–26 | | | 568 (95.3) |
| 27–28 | | | 28 (4.7) |
| **Level of Education** | | | |
| None or primary education | 839 (65.5) | 492 (63.6) | 385 (64.6) |
| Secondary or post-secondary education | 440 (34.3) | 280 (36.2) | 211 (35.4) |
| Missing | 2 (0.2) | 1 (0.1) | 0 |
| **Mothers' HIV status at time of study screening** (4–26 weeks post-partum) | | | |
| Already known HIV-infected | 1208 (94.3) | 743 (96.1) | 579 (97.1) |
| Newly diagnosed HIV-infected | 73 (5.7) | 30 (3.9) | 17 (2.9) |
| **Mothers' reported disclosure of her HIV status to her partner at each visit** | | | |
| Yes, partner knows her HIV-positive status | 1058 (82.6) | 651 (84.2) | 502 (84.2) |
| No, partner does not know her HIV-positive status | 143 (11.2) | 27 (3.5) | 57 (9.6) |
| No partner | 73 (5.7) | 89 (11.5) | 32 (5.4) |
| Missing | 7 (0.5) | 6 (0.8) | 5 (0.8) |
| **Mothers' ART Initiation[$] (as reported at enrolment)** | | | |
| Started ART pre-conception | 561 (43.8) | 354 (45.8) | 288 (48.3) |
| Started ART post-conception (during pregnany or post-partum) | 590 (46.1) | 362 (46.8) | 278 (46.6) |
| Not started ART | 87 (6.8) | 53 (6.9) | 29 (4.9) |
| Missing | 43 (3.4) | 4 (0.5) | 1 (0.2) |
| **Mothers' ART status[$]** | | | |

(*Continued*)

**Table 1.** (Continued)

| | N | Enrolment 1–6 months post-partum | Annual visit 1 12 months post-partum | Annual visit 2 24 months post-partum |
|---|---|---|---|---|
| | N | 1281 | 773 | 596 |
| On ART | | 1178 (92.0) | 755* (97.7) | 590** (99.0) |
| Not started ART | | 87 (6.8) | 1 (0.1) | 0 |
| Started but Stopped ART | | 14 (1.1) | 15 (1.9) | 6 (1.0) |
| Missing | | 2 (0.2) | 2 (0.3) | 0 |
| **Time on ART in months (at time of visit) among those on ART, median (IQR)** | | 8.3 (5.2–39.5) | 19.2 (15.9–50.4) | 30.1 (26.8–61.3) |
| **Time on ART in months (at time of visit) among those on ART** | | | | |
| ≤3.0 | | 75 (6.4) | 3 (0.4) | 0 |
| 3.1–6.0 | | 275 (23.3) | 3 (0.4) | 0 |
| 6.1–12.0 | | 229 (19.4) | 49 (6.5) | 4 (0.7) |
| 12.1–18.0 | | 31 (2.6) | 238 (31.5) | 6 (1.0) |
| 18.1–24.0 | | 39 (3.3) | 89 (11.8) | 47 (8.0) |
| ≥24 | | 345 (29.3) | 297 (39.3) | 483 (81.9) |
| Missing | | 184 (15.6) | 76 (10.1) | 50 (8.5) |
| **Mothers' self-reported health status** | | | | |
| Well | | 1214 (94.8) | 746 (96.5) | 563 (94.5) |
| Minor Illness | | 52 (4.1) | 23 (3.0) | 17 (2.9) |
| Major Illness | | 8 (0.6) | 0 | 0 |
| Missing | | 7 (0.5) | 4 (0.5) | 16 (2.7) |

$^\$$ Self-reported and verified/amended with clinical records when available.

* Including 39 women not (yet) on ART at enrolment of which 24 newly diagnosed at study enrolment.

** Including 27 women not (yet) on ART at enrolment of which 17 newly diagnosed at study enrolment.

enrollment experienced high loss to follow-up in the cohort (17/73 (23.3%) were retained to 24 months); however, the majority of women who started ART post-enrollment (15/17; 88.2%) gained VLS after ART initiation.

Table 4 describes factors associated with having a detectable VL at 24 months. In univariable analysis, both sub-optimal adherence measures and detectable VL in previous visits were associated with detectable VL at 24 months. In multivariable analysis (adjusted for age, parity, education, partner disclosure, timing of ART start and adherence), having detectable VL at 24 months was 9.0 times more likely among women with 1 prior detectable VL (95% CI 3.5–23.0, p<0.0001) and 226.4 times more likely for women with 2 prior detectable VLs (95% CI 73.0–701.8, p<0.0001).

Table 5A describes risk factors associated with experiencing any further viremia during the post-partum period among women who had undetectable VL at enrollment (N = 361). In multivariable analysis, having at least one sub-optimal adherence measure tripled the risk of a having a viremic episode during the 24 months post-partum (aOR 3.2, 95% CI 1.1–9.4, p = 0.03; controlled for age, parity, education, partner disclosure and timing of ART initiation).

Table 5B describes risk factors associated with having a persistent detectable VL through the post-partum period (vs. all others with at least one undetectable VL<40). In multivariable analysis, having at least one sub-optimal adherence measure more than doubled the risk of persistent viremia (aOR 2.3, 95% CI 1.1–4.9, p = 0.03; controlled for age, parity, education, partner disclosure and timing of ART initiation).

**Table 2. Adherence and viral load over time among mothers on ART.**

| | Enrolment | 12 months | 24 months |
|---|---|---|---|
| N* | 1178 | 755* | 590** |
| **Nr of days having missed ART in the last month, among mothers on ART, at each visit** | | | |
| 0 | 925 (78.5) | 645 (85.4) | 498 (84.4) |
| 1 day | 96 (8.1) | 51 (6.8) | 41 (6.9) |
| ≥2 days | 146 (12.4) | 57 (7.5) | 36 (6.1) |
| Missing | 11 (0.9) | 2 (0.3) | 15 (2.5) |
| **Combined self-reported adherence at 12 and 24 month** | | | |
| Fully optimal# adherence over time | - | 600 (79.5) | 477 (80.8) |
| At least one sub-optimal## adherence measure | - | 131 (17.4) | 104 (17.6) |
| Missing (data available for 1 visit only) | - | 24 (3.2) | 9 (1.5) |
| **Viral load at each visit** | | | |
| <40 | 912 (77.4) | 637 (84.4) | 473 (80.2) |
| 40–1000 | 90 (7.6) | 21 (2.8) | 19 (3.2) |
| >1000 | 145 (12.3) | 61 (8.1) | 48 (8.1) |
| Unknown/Missing | 31 (2.6) | 36 (4.8) | 50 (8.5) |
| **Nr of consecutive viral load observations** | | | |
| 0 | 31 (2.6) | 1 (0.1) | |
| 1 | 1147 (97.4) | 53 (7.0) | 8 (1.4) |
| 2 | | 701 (92.8) | 140 (23.7) |
| 3 | | | 442 (74.9) |
| **Cumulative detectable viral loads*** | n = 1147 | n = 701 | n = 442*** |
| 0 current and previous detectable (>40) VL | 912 (79.5) | 548 (78.2) | 345 (78.1) |
| 1 current or previous detectable (>40) VL | 235 (20.5) | 88 (12.6) | 48 (10.9) |
| 2 current and/or previous detectable (>40) VL | | 65 (9.3) | 16 (3.6) |
| 3 current and previous detectable (>40) VL | | | 33 (7.5) |

* Including 39 women reported not on ART at enrolment of which 24 newly diagnosed at study enrolment.

** Including 27 women reported not on ART at enrolment of which 17 newly diagnosed at study enrolment.

*** Including 17 women reported not on ART at enrolment of which 10 newly diagnosed at study enrolment.

# Reported to have missed ART 0 or 1 day in the last month in all visits (2 or 3).

## Reported to have missed ART 2 or more days in the last month in 1 or more previous visits.

## Discussion

We present long-term VLS data among Malawian women living with HIV (WLHIV) to 24 months post-partum. While we document high ART coverage (>93%) among women retained in the study and that optimal adherence was high (87–92%) when measured at each time point, the proportion of women with durable adherence throughout post-partum was lower (80.8%). Similarly, while the proportion of women with VLS at each time point met the UNAIDS goal of >90%, fewer women were able to maintain durable VLS throughout the post-partum period: approximately 20% of women experienced at least one episode of viremia and 7.5% had persistently detectable VL. Among women with complete VL data, detectable VL in the early post-partum period signaled an increased risk of ongoing viremia at both 12 and 24 months post-partum. Sub-optimal adherence was significantly associated with both losing viral suppression among women who had VLS at enrollment and with having persistently detectable VL.

Currently, limited data exist regarding long-term VLS among women enrolled in PMTCT programs in sub-Saharan Africa [3]. The VLS proportions measured here at early post-partum,

**Table 3. Viral load trends among women with complete viral load data.**

| Women on ART from Study Enrolment, n = 425 | | | | | | Women NOT on ART at study enrolment, n = 17 | | | | | |
|---|---|---|---|---|---|---|---|---|---|---|---|
| Enrolment | | 12-month visit | | 24-month visit | | Enrolment | | 12-month visit | | 24-month visit | |
| <40 | 359 | <40 | 353 | <40 | 343$ | <40 | 2# | <40 | 2 | <40 | 2 |
| | | | | 40–1000 | 5 | | | | | 40–1000 | |
| | | | | >1000 | 5 | | | | | >1000 | |
| | | 40–1000 | 3 | <40 | 2 | | | 40–1000 | | <40 | |
| | | | | 40–1000 | 1 | | | | | 40–1000 | |
| | | | | >1000 | 0 | | | | | >1000 | |
| | | >1000 | 3 | <40 | 1 | | | >1000 | | <40 | |
| | | | | 40–1000 | 1 | | | | | 40–1000 | |
| | | | | >1000 | 1 | | | | | >1000 | |
| 40–1000 | 27 | <40 | 18 | <40 | 15 | | | | | | |
| | | | | 40–1000 | 3 | | | | | | |
| | | | | >1000 | 0 | | | | | | |
| | | 40–1000 | 4 | <40 | 0 | | | | | | |
| | | | | 40–1000 | 2* | | | | | | |
| | | | | >1000 | 2* | | | | | | |
| | | >1000 | 5 | <40 | 1 | | | | | | |
| | | | | 40–1000 | 0* | | | | | | |
| | | | | >1000 | 4* | | | | | | |
| >1000 | 39 | <40 | 12 | <40 | 9 | >1000 | 15## | <40 | 12 | <40 | 11 |
| | | | | 40–1000 | 0 | | | | | 40–1000 | 1 |
| | | | | >1000 | 3 | | | | | >1000 | 0 |
| | | 40–1000 | 3 | <40 | 0 | | | 40–1000 | 1 | <40 | |
| | | | | 40–1000 | 1* | | | | | 40–1000 | |
| | | | | >1000 | 2* | | | | | >1000 | 1 |
| | | >1000 | 24 | <40 | 3 | | | >1000 | 2 | <40 | 2 |
| | | | | 40–1000 | 1* | | | | | 40–1000 | |
| | | | | >1000 | 20* | | | | | >1000 | |

$ Durable undetectable viral load.

\* Persistent viremia (total:32).

\# Known HIV-positive mothers (1 not started, 1 stopped ART).

\#\# 10 newly diagnosed not yet on ART and 5 known HIV-positive mothers (3 not started and 2 stopped ART).

12 and 24 months approximated UNAIDS 90-90-90 goals and are similar to VLS reported in other countries in the region [8, 9]. We additionally report that approximately 80% retained in this cohort had durable VLS to 24 months, which will confer increased benefit in preventing HIV transmission during breastfeeding and subsequent pregnancies, as well as preservation of health in mothers. However, the proportion of women achieving durable VLS in our study is higher than that reported in South Africa (70% (<50 copies/mL) at 12 months and 56% at 36–60 months (median 44 months post-partum)) [18] and in Kenya (67% (<1000 copies/mL) at 12months) [19]. These differences may be explained by diverse proportions of retention and loss to follow-up across these different settings, including that women retained to 24 months in our study may not reflect population-level VLS among all post-partum Malawian women.

However, despite comparatively higher rates of durable VLS over 24 months in this cohort, approximately 1 in 5 women still experienced at least one post-partum episode of viremia which likely has implications for MTCT, as well as the health of the mothers. While we cannot

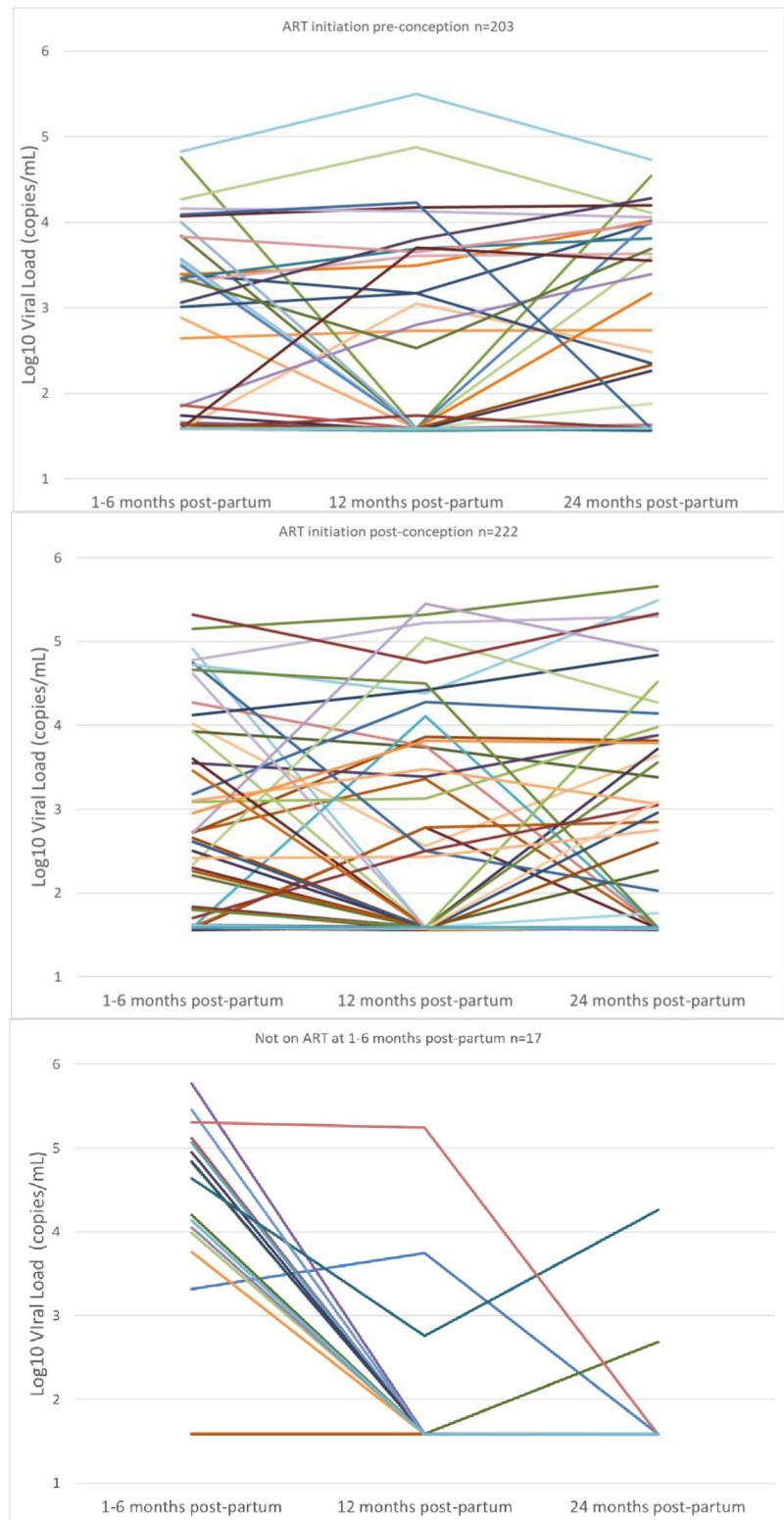

**Fig 2. Sequential viral load testing in a cohort of 442 mothers who were tested at 1–6, 12 and 24 months post-partum in Malawi stratified by ART start as either pre-conception, post-conception or post-enrollment.**

**Table 4. Factors associated with detectable viral load (VL >40) at 24 months post-partum among women on ART and having complete VL data (N = 442).**

| | n/n | % | Univariate (unadjusted) | | Multivariable (adjusted) | |
|---|---|---|---|---|---|---|
| | | | OR (95%CI) | p-value | aOR (95%CI)[#] | p-value |
| Total | 53/442 | 12.0 | | | | |
| **Mother's age in years** | | | | | | |
| ≤24 | 11/104 | 10.6 | 0.8 (0.4–1.9) | 0.60 | | |
| ≥25 | 42/336 | 12.5 | ref | | | |
| Missing | 0/2 | | | | | |
| **Parity, %** | | | | | | |
| 1–2 | 15/141 | 10.6 | 0.8 (0.4–1.5) | 0.55 | | |
| 3+ | 38/301 | 2.6 | ref | | | |
| **Level of Education** | | | | | | |
| None or primary education | 29/272 | 10.7 | 0.7 (0.4–1.3) | 0.28 | | |
| Secondary or post-secondary education | 24/170 | 14.1 | ref | | | |
| **Mother's reported disclosure of her HIV status to her partner at any time during the study** | | | | | | |
| Yes, partner knows her HIV-positive status | 48/400 | 12.0 | ref | | | |
| No partner throughout study period or mother never disclosed during study period | 5/42 | 12.2 | 1.0 (0.4–2.6) | 0.98 | | |
| **Maternal ART Initiation** | | | | | | |
| Pre-conception (started ART before last pregnancy) | 24/203 | 11.8 | ref | | | |
| Post-conception (started ART during last pregnancy or post-partum) | 27/222 | 12.2 | 1.0 (0.6–1.9) | 0.91 | | |
| New infections/not on ART at enrolment | 2/17 | 11.8 | 0.99 (0.2–4.6) | 0.99 | | |
| **Combined Self-reported adherence at 12 and 24 month (among those on ART)** | | | | | | |
| Fully optimal* adherence over time | 37/358 | 10.3 | ref | | | |
| At least one sub-optimal** adherence measure | 16/83 | 19.3 | 2.1 (1.1–3.9) | 0.03 | | |
| Missing (data available for 1 visit only) | 0/1 | 0.0 | | | | |
| **Cumulative detectable VLs** | | | | | | |
| 0 previous detectable (>40) VL | 10/355 | 2.8 | ref | | | |
| 1 previous detectable (>40) VL | 10/48 | 20.8 | 9.1 (3.6–23.2) | 0.0001 | 9.0 (3.5–23.0) | 0.0001 |
| 2 previous detectable (>40) VL | 33/39 | 84.6 | 189.8 (64.9–555.0) | 0.0001 | 226.4 (73.0–701.8) | 0.0001 |

[#] All variables were simultaneously entered in the logistic regression model as the first step and tested for removal one by one. In the multivariable analysis, only variables with significant associations in the last step are shown.

* Reported to have missed ART 0 or 1 day in the last month in all visits (2 or 3).

** Reported to have missed ART 2 or more days in the last month in 1 or more previous visits.

specifically comment on the duration or frequency of viremic episodes in relationship to individual level MTCT (as both were measured at yearly intervals over 24 months), we previously showed that non-suppressed VL, including low-level viremia (>40–1000 copies/ml) measured in the early post-partum period was predictive of MTCT at enrolment (4–26 weeks) in this same cohort of women [13]. Further, in Zimbabwe, a similar PMTCT program using a universal test and treat strategy linked non-durable VLS to increased MTCT risk to 18 months [8]. Considering this impact, and in light of emerging evidence here and in other studies in the region suggesting that women in pregnancy and post-partum may experience frequent episodes of viremia, this is an important finding for PMTCT programs [6].

What drives episodic viremia among post-partum women is likely to be challenges with adherence. Myer et al. [12] showed that ART non-adherence (versus drug-resistant mutations)

**Table 5. a and b Factors associated with experiencing any viremia during the post-partum period and with persistent detectable VL load through the post-partum period.**

| | a. Losing VLS over time (ie. not maintaining it). Those with at least one VL measure >40 vs those that had 3/3 VLS<40 | | | | | | b. Persistent viremia Those on ART at 24 months with 3/3 >40 vs all others (who had at least one VL<40) | | | | | |
|---|---|---|---|---|---|---|---|---|---|---|---|---|
| | n/n | % | Univariate (unadjusted) | | Multivariable (adjusted) | | n/n | % | Univariate (unadjusted) | | Multivariable (adjusted) | |
| | | | OR (95%CI) | p-value | aOR (95%CI)* | p-value | | | OR (95%CI) | p-value | aOR (95%CI)* | p-value |
| Total | 16/361 | 4.4 | | | | | 33/422 | 7.8 | | | | |
| **Mothers' age in years** | | | | | | | | | | | | |
| ≤24 | 5/80 | 6.3 | 1.6 (0.5–4.8) | 0.38 | | | 7/104 | 6.7 | 0.9 (0.4–2.0) | 0.73 | | |
| 25+ | 11/279 | 3.9 | ref | | | | 26/336 | 7.7 | ref | | | |
| Missing | | | | | | | 0/2 | | | | | |
| **Parity** | | | | | | | | | | | | |
| ≤2 | 7/117 | 6.0 | 1.7 (0.6–4.6) | 0.33 | | | 8/141 | 5.7 | 0.7 (0.3–1.5) | 0.33 | | |
| 3+ | 9/244 | 3.7 | ref | | | | 25/301 | 8.3 | ref | | | |
| **Level of Education, %** | | | | | | | | | | | | |
| None or Primary Education | 10/225 | 4.4 | 1.0 (0.4–3.0) | 0.99 | | | 17/272 | 6.3 | 0.6 (0.3–1.3) | 0.22 | | |
| Secondary and post-secondary education | 6/136 | 4.4 | ref | | | | 16/170 | 9.4 | ref | | | |
| **Mothers' reported disclosure of her HIV status to her partner at any time during the study** | | | | | | | | | | | | |
| Yes, partner knows her HIV-positive status | 15/326 | 4.6 | ref | | | | 29/400 | 7.3 | ref | | | |
| No partner throughout study period or mother never disclosed during study period | 1/35 | 2.9 | 0.6 (0.1–4.8) | 0.64 | | | 4/42 | 9.5 | 1.3 (0.4–4.0) | 0.59 | | |
| **Mothers' ART initiation** | | | | | | | | | | | | |
| Pre-conception (started ART before last pregnancy) | 6/176 | 3.4 | ref | | | | 14/203 | 6.9 | ref | | | |
| Post-conception (started ART during last pregnancy or post-partum) | 10/183 | 5.5 | 1.6 (0.6–4.6) | 0.35 | | | 18/222 | 8.1 | 1.2 (0.6–2.5) | 0.64 | | |
| New infections/not on ART (yet) at enrolment | 0/2 | 0 | | | | | 1/17 | 5.9 | 0.8 (0.1–6.8) | 0.87 | | |
| **Combined self-reported adherence at 12 and 24 month** | | | | | | | | | | | | |
| Fully optimal* adherence over time | 10/302 | 3.3 | ref | | ref | | 22/358 | 6.1 | ref | | Ref | |
| At least one sub-optimal** adherence measure | 6/59 | 10.2 | 3.3 (1.2–9.5) | 0.03 | 3.2 (1.1–9.4) | 0.03 | 11/83 | 13.3 | 2.3 (1.1–5.0) | 0.03 | 2.3 (1.1–4.9) | 0.03 |
| Missing (data available for 1 visit only) | | | | | | | 0/1 | | | | | |

* All variables were simultaneously entered in the logistic regression model as the first step and tested for removal one by one. In the multivariable analysis, only variables with significant associations in the last step are shown.

** Reported to have missed ART 0 or 1 day in the last month in all visits (2 or 3).

explain the vast majority of new viremic episodes in pregnant and post-partum women with elevated VL after initial suppression. [12] Our study adds evidence that maintaining durable adherence is difficult in this period, with more than 20% of women retained to 24 months

having at least one sub-optimal adherence measure [20, 21]. Literature has documented challenges in adherence, retention and achieving VLS in women initiating ART during pregnancy and breastfeeding [3, 7, 10] and highlights the role of post-partum physical, emotional and life role changes, including the increased care demands of an infant, as presenting particular challenges for this population [18, 20, 22]. Further studies highlight that non-adherence in pregnancy and the post-partum period is additionally related to biomedical, individual and health system-related factors, such as ART toxicity, comorbidities, stigma, HIV-status disclosure, mental health, food insufficiency, healthcare worker attitudes and the availability of supportive services [23–32].

While myriad interventions have been studied to support adherence, and subsequent loss of viral control in WLHIV in pregnancy and breastfeeding, finding ways to specifically identify women most at risk may be of benefit to programs to enable early targeted or differentiated care. Phillips et al. show that measures of self-reported adherence repeated over time are effective in identifying both current or pending elevated VL in HIV-infected pregnant and post-partum populations on ART [7, 18]. Further, we show that one VL measure early in the post-partum period strongly signals a risk of ongoing VL non-suppression up to 24 months. Currently, few countries implement a strategy of intensified VL monitoring during pregnancy, delivery, and post-partum, partly due to limited operational capacity and lack of a global consensus on best practices for VL testing during pregnancy and breastfeeding [33], however these results suggest that targeting at least one VL early in the post-partum period would identify women with high VL who are at highest risk of non-suppression through post-partum. The results of our study underline that further research is needed to determine the optimal timing and frequency of VL monitoring among pregnant and breastfeeding women to identify at-risk women and effective interventions that can achieve durable VLS in order to reach virtual elimination of MTCT.

## Strengths and limitations

The strengths of this study are in estimating levels of VL suppression at three time points in the post-partum period within a nationally representative cohort as part of a programmatic evaluation of Option B+ in Malawi. However, loss to follow-up in this cohort was high over 24 months which may bias our results. While a portion of this analysis included a smaller subset of only women with complete data over 24 months, the characteristics of this sub-cohort as described in the results likely lead to an underestimation of the number and proportion of viremic episodes and resulting implications for MTCT rates. We showed that detectable VL at enrollment conferred significant risk of continuing viremia at 12 and 24 months among those retained in the study. The risk of ongoing viremia is likely higher in those lost to follow (who were younger with lower parity and were more likely to have detectable VL at enrolment) and subsequently in the WLHIV population. The NEMAPP study deliberately did not place dedicated study staff at health facilities as it aimed to evaluate routine circumstances in the field. This resulted in regularly missed study procedures during routine clinic visits. Ministry of Health staff, burdened with a heavy routine clinic workload, also had to conduct study defaulter tracing activities, which is likely to have led to higher loss-to follow up rates. Loss to follow-up was high but anticipated in the original sample size calculations. Finally, we did not examine the potential role of drug resistance.

## Conclusion

Detectable VL early post-partum strongly increases risk of ongoing post-partum viremia with implications for infant HIV transmission, as well as for the health of the mother. While the

proportion of women retained in the study met the UNAIDS 90-90-90 goals of >90% VLS, a significantly lower proportion achieved durable sustained VLS. Due to mothers with non-sustained VLS during the breastfeeding period, coupled with the likelihood of following pregnancies in a high fertility setting, large numbers of infants remain at risk of acquiring HIV infection. Additionally, with UNAIDS recently releasing new targets for 95% of pregnant and breastfeeding women to reach viral suppression to achieve the elimination of transmission [34], intensified and differentiated VL monitoring and targeted adherence support is required both during pregnancy and the breastfeeding period.

## Supporting information

**S1 Table. Baseline characteristics of women who completed the study versus those lost-to-follow from study.**
(DOCX)

**S1 Data.**
(XLSX)

**S1 File.**
(PDF)

## Acknowledgments

We would like to acknowledge Dr. Marie Louise Newell for her contributions to this manuscript.

## Author Contributions

**Conceptualization:** Megan Landes, Monique van Lettow, Joep J. van Oosterhout, Erik Schouten, Beth A. Tippett Barr.

**Formal analysis:** Monique van Lettow.

**Investigation:** Megan Landes, Monique van Lettow, Joep J. van Oosterhout, Beth A. Tippett Barr.

**Methodology:** Megan Landes, Monique van Lettow, Joep J. van Oosterhout, Erik Schouten, Andrew Auld, Thokozani Kalua, Andreas Jahn, Beth A. Tippett Barr.

**Validation:** Megan Landes, Andrew Auld, Thokozani Kalua, Andreas Jahn, Beth A. Tippett Barr.

**Writing – original draft:** Megan Landes, Monique van Lettow, Beth A. Tippett Barr.

**Writing – review & editing:** Megan Landes, Monique van Lettow, Joep J. van Oosterhout, Erik Schouten, Andrew Auld, Thokozani Kalua, Andreas Jahn, Beth A. Tippett Barr.

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
