## [Decision Letter · Decision Letter 0]

15 Dec 2020

PONE-D-20-36329

Early post-partum viremia predicts long-term non-suppression of viral load in HIV-positive women on ART in Malawi : Implications for the elimination of infant transmission

PLOS ONE

Dear Dr. van Lettow,

Thank you for submitting your manuscript to PLOS ONE. After careful consideration, we feel that it has merit but does not fully meet PLOS ONE’s publication criteria as it currently stands. Therefore, we invite you to submit a revised version of the manuscript that addresses the points raised during the review process.

Please take into account all the pertinent comments raised by both reviewers when preparing the revised manuscript and verify that the statistical analysis of the data has been correctly performed.

We look forward to receiving your revised manuscript.

Kind regards,

Graciela Andrei

Academic Editor

PLOS ONE

Journal Requirements:

2. Please include additional information regarding the survey or questionnaire used in the study and ensure that you have provided sufficient details that others could replicate the analyses. For instance, if you developed a questionnaire as part of this study and it is not under a copyright more restrictive than CC-BY, please include a copy, in both the original language and English, as Supporting Information, or include a citation if it has been published previously.

3. In the Methods, please discuss whether and how the questionnaire was validated and/or pre-tested. If these did not occur, please provide the rationale for not doing so.

4. In statistical methods, please refer to any post-hoc corrections to correct for multiple comparisons during your statistical analyses. If these were not performed please justify the reasons. Please refer to our statistical reporting guidelines for assistance (https://journals.plos.org/plosone/s/submission-guidelines.#loc-statistical-reporting).

5. Please include a caption for figure 2.

6. We note that Supporting Information S2 Data with your paper includes detailed descriptions of individual patients/participants, i.e. data of birth.

As per the PLOS ONE policy (http://journals.plos.org/plosone/s/submission-guidelines#loc-human-subjects-research) on papers that include identifying, or potentially identifying, information, the individual(s) or parent(s)/guardian(s) must be informed of the terms of the PLOS open-access (CC-BY) license and provide specific permission for publication of these details under the terms of this license.

Please download the Consent Form for Publication in a PLOS Journal (http://journals.plos.org/plosone/s/file?id=8ce6/plos-consent-form-english.pdf). The signed consent form should not be submitted with the manuscript, but should be securely filed in the individual's case notes.

Please amend the methods section and ethics statement of the manuscript to explicitly state that the patient/participant has provided consent for publication: “The individual in this manuscript has given written informed consent (as outlined in PLOS consent form) to publish these case details”.

Reviewers' comments:

Reviewer's Responses to Questions

**Comments to the Author**

1. Is the manuscript technically sound, and do the data support the conclusions?

Reviewer #1: Yes

Reviewer #2: Yes

2. Has the statistical analysis been performed appropriately and rigorously? 

Reviewer #1: No

Reviewer #2: Yes

3. Have the authors made all data underlying the findings in their manuscript fully available?

Reviewer #1: Yes

Reviewer #2: Yes

4. Is the manuscript presented in an intelligible fashion and written in standard English?

Reviewer #1: Yes

Reviewer #2: Yes

5. Review Comments to the Author

Reviewer #1: The authors present a well described analysis of post-partum HIV viral loads among women on ART in Malawi. Overall the analysis provides a good report from a setting that has had a recent significant scale up in viral load testing in a key population for the elimination of MTCT. I have only one major issue, which is surrounding the approach to the multivariable analysis, otherwise there are a few points that could be clarified.

Major:

Methods, lines 77. PPS sampling is implied here, but I don't think the analysis is weighted for this - this was not clear to me.

Methods, lines 116-119. Stepwise selection is known to be a historically bad method of variable selection. It has many issues and does not represent a sound method of reducing/selecting variables for consideration. This is the reason that I indicated the analysis was not correct. Usually when interested in associations or risk factors we should have considered carefully the potential exposures/risk factors, as well as measured confounders, and other adjustment factors and should estimate our model from that basis. In a strictly predictive setting, again, stepwise selection would be the incorrect approach - other methods such as penalisation would perform better, but also in a strictly predictive setting we would evaluate our model at least under some sort of internal validation process (such as k-fold cross validation).

Results:

The relatively small sample sizes and large OR & CI led me to question the stability of the estimates. Was any assessment for multi-collinearity and/or quasi-separation done?

This analysis primarily reports VLS among individuals retained - this needs to be emphasised much more in the discussion as it is very key to understanding the value of the analysis.

Table 3: Proportions might be useful here, and the authors could consider a flow diagram / alluvial type plot

Table 4 (and Table 5): It is unclear to me in the adjusted model if NS means not included, or not-significant. If the latter, the OR and CI intervals should be reported.

Minor comments:

Abstract: Please report OR estimates along with CI intervals (rather than just "9 times" etc)

Methods, lines 78-90: This is just not clear and I wonder if it could be rephrased so the sample size calculation for this can be better understood.

Reviewer #2: This is a well-written paper that presents important findings on postpartum viral load outcomes in a nationally representative sample of women living with HIV in Malawi. While the study reassuringly finds higher proportions of viral suppression than reported in other settings, these still remain below global targets and this study clearly shows that women who have one raised viral load remain at risk for subsequent viremia.

Overall, the paper is very clearly presented. I have a few comments for clarity.

1. I found the wording of the first sentence f the results section in the abstract to be confusing - can you revise to be more clear that this is comparing retained in study to lost from study- were 46.5% retained and older, or just retained?

2. I appreciate that the results have been clearly presented as being among women with completed viral loads. I do think that the point raised in lines 139-141 that women who were lost from the study largely were known to be alive and on ART through routine data is important. I think the authors could make this clearer perhaps by adding this into the Figure 1 flow diagram (lost from study but known to be in care) and also in the discussion around considering the impact of missing data. I agree that viremia if anything is underestimated but the fact that most women lost from the study were still in ART care is useful to support why other missing data methods have not been used.

3. For the models in Table 4 and 5, it’s not clear if the variables reported as NS in the adjusted models were included in the final model but results not reported as not significant, or if these variables were kicked out of the model in the backward stepwise process. I suggest the adjusted OR and 95% CI be reported for each variable that was in included in the final model following the stepwise selection rather than reporting as NS.

4. Do the authors have any thoughts on what could be done to best use the viral load information when it is available? When a woman has a raised VL postpartum she is at increased risk of either staying viremic or having subsequent viremia – what sort of interventions could be implemented at that time?

5. You may consider referencing the latest UNAIDS targets (in the 2020 world AIDS report) which include targets for viral suppression during breastfeeding (95% of pregnant and breastfeeding women living with HIV have suppressed viral loads). https://www.unaids.org/sites/default/files/media_asset/prevailing-against-pandemics_en.pdf

6. PLOS authors have the option to publish the peer review history of their article (what does this mean?). If published, this will include your full peer review and any attached files.

Reviewer #1: No

Reviewer #2: No

---

## [Author Response · Author response to Decision Letter 0]

4 Jan 2021

PONE-D-20-36329

Early post-partum viremia predicts long-term non-suppression of viral load in HIV-positive women on ART in Malawi: Implications for the elimination of infant transmission

Responses to reviewers’ comments:

Response: We have verified that the manuscript meets the requirements and made a few adjustments as needed 

2. Please include additional information regarding the survey or questionnaire used in the study and ensure that you have provided sufficient details that others could replicate the analyses. For instance, if you developed a questionnaire as part of this study and it is not under a copyright more restrictive than CC-BY, please include a copy, in both the original language and English, as Supporting Information, or include a citation if it has been published previously.

Response: We have included the relevant questionnaires (that include both English and the local language) as supporting information. Please note that only a selection of questions from these questionnaires were used for this sub-analysis. 

3. In the Methods, please discuss whether and how the questionnaire was validated and/or pre-tested. If these did not occur, please provide the rationale for not doing so.

Response: questionnaires were pre-tested prior to study start. This has now been included in the method section. 

4. In statistical methods, please refer to any post-hoc corrections to correct for multiple comparisons during your statistical analyses. If these were not performed please justify the reasons. Please refer to our statistical reporting guidelines for assistance (https://journals.plos.org/plosone/s/submission-guidelines.#loc-statistical-reporting).

Response: We believe that we described all the technical details and procedures required to reproduce the analysis. Post-hoc corrections were not performed/not deemed necessary. The statistical methods were reviewed and approved by 2 statisticians during CDC internal review. 

5. Please include a caption for figure 2.

 Response: thanks, this has now been included

6. We note that Supporting Information S2 Data with your paper includes detailed descriptions of individual patients/participants, i.e. data of birth.

As per the PLOS ONE policy (http://journals.plos.org/plosone/s/submission-guidelines#loc-human-subjects-research) on papers that include identifying, or potentially identifying, information, the individual(s) or parent(s)/guardian(s) must be informed of the terms of the PLOS open-access (CC-BY) license and provide specific permission for publication of these details under the terms of this license.

Please download the Consent Form for Publication in a PLOS Journal (http://journals.plos.org/plosone/s/file?id=8ce6/plos-consent-form-english.pdf). The signed consent form should not be submitted with the manuscript, but should be securely filed in the individual's case notes.

Please amend the methods section and ethics statement of the manuscript to explicitly state that the patient/participant has provided consent for publication: “The individual in this manuscript has given written informed consent (as outlined in PLOS consent form) to publish these case details”.

Response: We belief that without any other data provided, participants cannot be identified through the date of birth only. However, we have taken out date of birth in the dataset. Please delete the current version from your depository. 

Reviewers' comments:

Reviewer's Responses to Questions

Comments to the Author

1. Is the manuscript technically sound, and do the data support the conclusions?

Reviewer #1: Yes

Reviewer #2: Yes

2. Has the statistical analysis been performed appropriately and rigorously? 

Reviewer #1: No

Reviewer #2: Yes

3. Have the authors made all data underlying the findings in their manuscript fully available?

Reviewer #1: Yes

Reviewer #2: Yes

4. Is the manuscript presented in an intelligible fashion and written in standard English?

Reviewer #1: Yes

Reviewer #2: Yes

5. Review Comments to the Author

Reviewer #1: The authors present a well described analysis of post-partum HIV viral loads among women on ART in Malawi. Overall, the analysis provides a good report from a setting that has had a recent significant scale up in viral load testing in a key population for the elimination of MTCT. I have only one major issue, which is surrounding the approach to the multivariable analysis, otherwise there are a few points that could be clarified.

Major:

Methods, lines 77. PPS sampling is implied here, but I don't think the analysis is weighted for this - this was not clear to me.

Response: Thank you, this was not correctly described and has now been amended. 

PPS sampling was for the total cohort. Indeed, for the total cohort we used weighted analyses to control for the survey design. However, for this sub-set weighted analysis would not be appropriate. 

Methods, lines 116-119. Stepwise selection is known to be a historically bad method of variable selection. It has many issues and does not represent a sound method of reducing/selecting variables for consideration. This is the reason that I indicated the analysis was not correct. Usually when interested in associations or risk factors we should have considered carefully the potential exposures/risk factors, as well as measured confounders, and other adjustment factors and should estimate our model from that basis. In a strictly predictive setting, again, stepwise selection would be the incorrect approach - other methods such as penalisation would perform better, but also in a strictly predictive setting we would evaluate our model at least under some sort of internal validation process (such as k-fold cross validation).

Response: 

Thank you for your careful consideration of the analysis plan. We utilized the stepwise selection after discussion with CDC statisticians. After consulting again with them on receiving your review feedback, they advised that while from an academic standpoint it may not be ideal, the disadvantages are not considered ‘bad’ or ‘unsound’ as it remains a very practical approach in this context. In our early infant transmission (6-12wks of age) analysis conducted with study enrolment data, we developed and utilized a directed acyclic graph (DAG) to identify confounders and guide selection of variables (see Lancet article embedded and figure pasted below). These same variables were therefore most likely to be relevant in the model utilized at 24mos of age, and because we were able to use existing knowledge to a-priori determine what factors would be reasonable to include in the model, then the method chosen was appropriate. 

If preferred, we would be happy to reference or reproduce this DAG in this paper as well, as the variables included are the same. Thank you in advance for your guidance.

Results:

The relatively small sample sizes and large OR & CI led me to question the stability of the estimates. Was any assessment for multi-collinearity and/or quasi-separation done?

Response: We agree that this final model represents a large OR and CI based on the relatively small sample size. We did not employ the methods suggested but have ensured that we are highlighting this in the limitations of our analysis. 

This analysis primarily reports VLS among individuals retained - this needs to be emphasised much more in the discussion as it is very key to understanding the value of the analysis.

Response: we have emphasized this further in the discussion, as suggested. 

Table 3: Proportions might be useful here, and the authors could consider a flow diagram / alluvial type plot

Response: We have tried several versions of charts but concluded that the current (sideways) flow chart is clearest. As there are several levels of comparisons, we did not complete proportions. These are described in the text. 

Table 4 (and Table 5): It is unclear to me in the adjusted model if NS means not included, or not-significant. If the latter, the OR and CI intervals should be reported.

Response: Thank you – this was indeed not clear and has now been amended in tables and footnotes. “All variables were simultaneously entered in the logistic regression model as the first step and tested for removal one by one. In the multivariable analysis, only variables with significant associations in the last step are shown.” 

Minor comments:

Abstract: Please report OR estimates along with CI intervals (rather than just "9 times" etc)

Response: The aORs have been added

Methods, lines 78-90: This is just not clear and I wonder if it could be rephrased so the sample size calculation for this can be better understood.

Response: this has now been rephrased. 

Reviewer #2: This is a well-written paper that presents important findings on postpartum viral load outcomes in a nationally representative sample of women living with HIV in Malawi. While the study reassuringly finds higher proportions of viral suppression than reported in other settings, these still remain below global targets and this study clearly shows that women who have one raised viral load remain at risk for subsequent viremia.

Overall, the paper is very clearly presented. I have a few comments for clarity.

1. I found the wording of the first sentence of the results section in the abstract to be confusing - can you revise to be more clear that this is comparing retained in study to lost from study- were 46.5% retained and older, or just retained?

Response: this has now been made more clear

2. I appreciate that the results have been clearly presented as being among women with completed viral loads. I do think that the point raised in lines 139-141 that women who were lost from the study largely were known to be alive and on ART through routine data is important. I think the authors could make this clearer perhaps by adding this into the Figure 1 flow diagram (lost from study but known to be in care) and also in the discussion around considering the impact of missing data. I agree that viremia if anything is underestimated but the fact that most women lost from the study were still in ART care is useful to support why other missing data methods have not been used.

Response: Thank you – that is a good idea. We have now added those lost n study but know to be in care in Figure 1. The impact of missing data has also been further discussed in the Discussion 

3. For the models in Table 4 and 5, it’s not clear if the variables reported as NS in the adjusted models were included in the final model but results not reported as not significant, or if these variables were kicked out of the model in the backward stepwise process. I suggest the adjusted OR and 95% CI be reported for each variable that was in included in the final model following the stepwise selection rather than reporting as NS. 

Response: the aORs and 95% CI have now been added to the tables for those that were not significant but included in the model.

4. Do the authors have any thoughts on what could be done to best use the viral load information when it is available? When a woman has a raised VL postpartum she is at increased risk of either staying viremic or having subsequent viremia – what sort of interventions could be implemented at that time?

Response: We have added our suggestion to the results – that where resources are limited for VL, targeting at least one in the PP period would allow for identification of those women at highest risk of non-suppression throughout PP. (Ideally this VL would be during pregnancy as well, but this is beyond the scope/implications of our PP findings). 

5. You may consider referencing the latest UNAIDS targets (in the 2020 world AIDS report) which include targets for viral suppression during breastfeeding (95% of pregnant and breastfeeding women living with HIV have suppressed viral loads). https://www.unaids.org/sites/default/files/media_asset/prevailing-against-pandemics_en.pdf

Response: Thank you. This has been added as suggested

6. PLOS authors have the option to publish the peer review history of their article (what does this mean?). If published, this will include your full peer review and any attached files.

Do you want your identity to be public for this peer review? For information about this choice, including consent withdrawal, please see our Privacy Policy.

Reviewer #1: No

Reviewer #2: No

---

## [Decision Letter · Decision Letter 1]

26 Jan 2021

PONE-D-20-36329R1

Early post-partum viremia predicts long-term non-suppression of viral load in HIV-positive women on ART in Malawi : Implications for the elimination of infant transmission

PLOS ONE

Dear Dr. van Lettow,

Thank you for submitting your manuscript to PLOS ONE. After careful consideration, we feel that it has merit but does not fully meet PLOS ONE’s publication criteria as it currently stands. Therefore, we invite you to submit a revised version of the manuscript that addresses the points raised during the review process.

As suggested by Reviewer # 1, when preparing the second revised version, please remove from the manuscript the multivariable results, which are not relevant and actually misleading.

We look forward to receiving your revised manuscript.

Kind regards,

Graciela Andrei

Academic Editor

PLOS ONE

Reviewers' comments:

Reviewer's Responses to Questions

**Comments to the Author**

1. If the authors have adequately addressed your comments raised in a previous round of review and you feel that this manuscript is now acceptable for publication, you may indicate that here to bypass the “Comments to the Author” section, enter your conflict of interest statement in the “Confidential to Editor” section, and submit your "Accept" recommendation.

Reviewer #1: (No Response)

Reviewer #2: All comments have been addressed

2. Is the manuscript technically sound, and do the data support the conclusions?

Reviewer #1: Yes

Reviewer #2: Yes

3. Has the statistical analysis been performed appropriately and rigorously? 

Reviewer #1: No

Reviewer #2: Yes

4. Have the authors made all data underlying the findings in their manuscript fully available?

Reviewer #1: Yes

Reviewer #2: Yes

5. Is the manuscript presented in an intelligible fashion and written in standard English?

Reviewer #1: Yes

Reviewer #2: Yes

6. Review Comments to the Author

Reviewer #1: The analysis is not acceptable to me while there is inference made on a multivariable method where variable selection is by step wise regression was carried out. The multivariable analysis could be removed, and I would be happy to recommend the manuscript for publication.

Issues surrounding step wise selection of variables in multivariable regression include having biased upward R2 and biased downward standard errors where F and chi-2 test statistics do not have claimed distribution - this means the CI intervals around the parameter estimates are incorrect. The p-values are too low (and in fact not p-values as we understand them) because of multiple comparisons, but very difficult to correct because of the dependent nature of the steps. Parameter estimates tend to be biased high (in absolute value), and there is no guarantee that any or all of the variables with 'true' associations with the outcome will be identified.

Relevant references:

Altman DG, Andersen PK. 1989. Bootstrap investigation of the stability of a Cox regression model. Statistics in Medicine 8: 771–783.

Burnham KP, Anderson DR. (2002), Model selection and multimodel inference, Springer, New York.

Copas JB. 1983. Regression, prediction and shrinkage (with discussion). Journal of the Royal Statistical Society, Series B 45: 311–354.

Derksen S. Keselman HJ. 1992. Backward, forward and stepwise automated subset selection algorithms: frequency of obtaining authentic and noise variables. British Journal of Mathematical and Statistical Psychology 45: 265–282.

Harrell F E (2001), Regression modeling strategies: With applications to linear models, logistic regression, and survival

analysis, Springer-Verlag, New York.

Hurvich CM, Tsai CL. 1990. The impact of model selection on inference in linear regression. American Statistician 44: 214–217.

Judd, McClelland. Data Analysis: A Model Comparison Approach (Harcourt Brace Jovanovich, ISBN 0-15-516765-0)

Mantel N. 1970. Why stepdown procedures in variable selection. Technometrics 12: 621–625.

Roecker EB. 1991. Prediction error and its estimation for subset—selected models. Technometrics 33: 459–468.

Tibshirani R. 1996. Regression shrinkage and selection via the lasso. Journal of the Royal Statistical Society, Series B 58: 267–288.

Reviewer #2: (No Response)

7. PLOS authors have the option to publish the peer review history of their article (what does this mean?). If published, this will include your full peer review and any attached files.

Reviewer #1: No

Reviewer #2: No

---

## [Author Response · Author response to Decision Letter 1]

25 Feb 2021

PONE-D-20-36329R1

Early post-partum viremia predicts long-term non-suppression of viral load in HIV-positive women on ART in Malawi : Implications for the elimination of infant transmission

PLOS ONE

Response to comment Reviewer 1: 

Reviewer # 1, when preparing the second revised version, please remove from the manuscript the multivariable results, which are not relevant and actually misleading.

Thank you for your thorough review. 

We consulted with academic and programmatic statisticians and discussed the feedback received.

As described in the former response to reviewers’ comments we were advised that while from an academic standpoint our analyses may not be ideal, that the disadvantages were not considered ‘bad’ or ‘unsound’ as it remains a very practical approach in this context. We also explained that in our early infant transmission (6-12wks of age) analysis conducted with study enrolment data, we developed and utilized a directed acyclic graph (DAG) to identify confounders and guide selection of variables (see Lancet article: https://www.thelancet.com/pdfs/journals/lanhiv/PIIS2352-3018(18)30316-3.pdf). These same variables were therefore most likely to be relevant in the model utilized at 24mos of age, and because we were able to use existing knowledge to a-priori determine what factors would be reasonable to include in the model, then the method chosen was appropriate. 

After careful consideration, we did not remove the multivariable results from the manuscript, as we disagree that the nuanced method changes would result in a measurable difference to the findings, and the paper would ultimately be of lower quality. As pragmatic researchers with a focus on creating evidence for programmatic improvement, we believe that our conclusions that women with non-suppressed viral load at early postpartum are likely to remain elevated at 12 months and 24 months needs to be published (ie. a single VL measure at this timepoint can identify the majority of women who remain unsuppressed).

---

## [Editor Report · Decision Letter 2]

2 Mar 2021

Early post-partum viremia predicts long-term non-suppression of viral load in HIV-positive women on ART in Malawi : Implications for the elimination of infant transmission

PONE-D-20-36329R2

Dear Dr. van Lettow,

We’re pleased to inform you that your manuscript has been judged scientifically suitable for publication and will be formally accepted for publication once it meets all outstanding technical requirements.

Kind regards,

Graciela Andrei

Academic Editor

PLOS ONE
---

## [Editor Report · Acceptance letter]

3 Mar 2021

PONE-D-20-36329R2 

Early post-partum viremia predicts long-term non-suppression of viral load in HIV-positive women on ART in Malawi: Implications for the elimination of infant transmission 

Dear Dr. van Lettow:

I'm pleased to inform you that your manuscript has been deemed suitable for publication in PLOS ONE. Congratulations! Your manuscript is now with our production department. 

Kind regards, 

on behalf of

Dr. Graciela Andrei 

Academic Editor

PLOS ONE